# Do Animals Play a Role in the Transmission of Severe Acute Respiratory Syndrome Coronavirus-2 (SARS-CoV-2)? A Commentary

**DOI:** 10.3390/ani11010016

**Published:** 2020-12-24

**Authors:** Anna Costagliola, Giovanna Liguori, Danila d’Angelo, Caterina Costa, Francesca Ciani, Antonio Giordano

**Affiliations:** 1Department of Veterinary Medicine and Animal Productions, University of Napoli Federico II, 80137 Napoli, Italy; costagli@unina.it (A.C.); danila.dangelo@unina.it (D.d.); ciani@unina.it (F.C.); 2Department of Prevention, ASL Foggia, 71122 Foggia, Italy; 3Cell Biology and Biotherapy Unit, Istituto Nazionale Tumori-IRCCS-Fondazione G. Pascale, 80131 Napoli, Italy; c.costa@istitutotumori.na.it; 4Sbarro Institute for Cancer Research and Molecular Medicine and Center of Biotechnology, College of Science and Technology, Temple University, Philadelphia, PA 19122, USA; giordano@temple.edu; 5Department of Medical Biotechnologies, University of Siena, 53100 Siena, Italy

**Keywords:** SARS-CoV-2, COVID-19, zoonosis, coronavirus

## Abstract

**Simple Summary:**

This commentary addresses the zoonotic and epidemiological aspects of Severe acute respiratory syndrome coronavirus-2 (SARS-Cov-2) pandemic that is affecting the whole world with a contagion curve that continues to grow. This work aims to highlight the role that animals might play in the epidemiology of the disease. As knowledge of SARS-CoV-2 has progressed, most of the scientific information confirms that animals cannot transmit the virus to humans. There is evidence that people who have tested positive for COVID-19 can infect pets, farm animals, and wild animals. However, very recently, a SARS-Cov-2 variant related to minks has been found in mink workers in Denmark with a disease severity that is similar to those infected with non-mink-related variants. Further studies are therefore necessary. The concept of “One Health” recognizes the interdependence of human, animal, and environmental health, and aims to improve public health outcomes through the understanding and prevention of risks that originate at the interfaces between humans, animals, and their environments. It is therefore important that veterinarians share information with public health officers to assess the risks of transmission from people infected with COVID-19 to animals, or to determine when animals could spread pandemic viruses.

**Abstract:**

Severe acute respiratory syndrome coronavirus-2 (SARS-CoV-2) belongs to the Beta-coronavirus genus. It is 96.2% homologous to bat CoV RaTG13 and 88% homologous to two bat SARS-like coronaviruses. SARS-CoV-2 is the infectious agent responsible for the coronavirus disease (COVID-19), which was first reported in the Hubei province of Wuhan, China, at the beginning of December 2019. Human transmission from COVID-19 patients or incubation carriers occurs via coughing, sneezing, speaking, discharge from the nose, or fecal contamination. Various strains of the virus have been reported around the world, with different virulence and behavior. In addition, SARS-CoV-2 shares certain epitopes with some taxonomically related viruses, with tropism for the most common synanthropic animals. By elucidating the immunological properties of the circulating SARS-CoV-2, a partial protection due to human–animal interactions could be supposed in some situations. In addition, differential epitopes could be used for the differential diagnosis of SARS-CoV-2 infection. There have been cases of transmission from people with COVID-19 to pets such as cats and dogs. In addition, wild felines were infected. All These animals were either asymptomatic or mildly symptomatic and recovered spontaneously. Experimental studies showed cats and ferrets to be more susceptible to COVID-19. COVID-19 positive dogs and felines do not transmit the infection to humans. In contrast, minks at farms were severely infected from people with COVID-19. A SARS-Cov-2 variant in the Danish farmed mink that had been previously infected by COVID-19 positive workers, spread to mink workers causing the first case of animal-to-human infection transmission that causes a moderate decreased sensitivity to neutralizing antibodies. Thus, more investigations are necessary. It remains important to understand the risk that people with COVID-19 pose to their pets, as well as wild or farm animals so effective recommendations and risk management measures against COVID-19 can be made. A One Health unit that facilitates collaboration between public health and veterinary services is recommended.

## 1. Introduction

Infectious diseases affect people, domestic animals, and wildlife, and several pathogens can infect multiple species. In the last few decades, there has been an increase in the number of emerging infectious zoonotic diseases in humans that originated from wildlife. Multiple factors such as globalization and urbanization (including mobility, human population density, contact structures, food production and consumption, deforestation and pollution, changes in climate, molecular evolutions in different host species, host responses to infections, and interactions between wild animals, domestic animals and humans, give pathogens opportunities to spread out, making prevention activities more difficult (Editorial. Emerging zoonoses). Coronaviruses (CoVs) are important pathogens for humans and vertebrates. They can infect the respiratory, gastrointestinal, hepatic, and central nervous systems of humans, avians, camels, bats, masked palm civets, mice, dogs, and cats [1].

CoVs are single-strand, ribonucleic acid (RNA) viruses with around 30 kilobases. They belong to the Coronaviridae family and include four genera: Alpha-, Beta-, Gamma-, and Delta-coronavirus. The Beta-coronavirus genus includes HCoV-OC43, HCoV-HKU1, severe acute respiratory syndrome coronavirus (SARS-CoV), Middle East respiratory syndrome-related coronavirus (MERS-CoV), and most recently severe acute respiratory disease coronavirus 2 (SARS-CoV-2) [2,3]. MERS-CoV and SARS-CoV cause zoonotic syndromes that can be transmitted from animal to human and from human to human. Both cause severe respiratory symptoms that can be fatal in humans [4]. In late December 2019, a novel CoV infection appeared in the Hubei province of Wuhan, China. It was named COVID-19 (corona-virus-disease-19) [5] and it rapidly spread around the world. In March 2020, COVID-19 was declared pandemic by the World Health Organization (WHO), due to its rapid diffusion from one continent to another. The International Committee on Taxonomy named SARS-CoV-2 as the virus responsible for COVID-19.

Phylogenetic analysis of the SARS-CoV-2 genome showed it belongs to the Beta-coronavirus genus, subgenus Sarbecovirus [6] and shares several similarities with other CoVs. Within the Beta-coronavirus genus, SARS-CoV-2 shares about 79% of its homology with the SARS-CoV that was transmitted to humans from civets causing the 2002–2003 epidemic [6,7]. Both use the same cell entry receptor, the angiotensin-converting enzyme II (ACE2) that is expressed on the membranes of the lungs, cardiovascular system, gut, kidney, central nervous system, and adipose tissue [8,9,10]. SARS-CoV-2 shares about 50% homology with MERS-CoV, which was transmitted to humans from dromedary camels and was responsible for the 2012 epidemic [6]. In contrast, SARS-CoV-2 shares 96.2% homology with bat CoV RaTG13 [11] and 88% homology with two bat-derived SARS-like coronaviruses, bat-SL-CoVZC45 and bat-SLCoVZXC21 [6]. Thus, it has been hypothesized that SARS-CoV-2 can be transmitted from a bat after mutation, thus conferring ability to infect humans [12,13]. However, bats are not sold in local Chinese markets [9]. Controversial results have been reported, suggesting the snake as an intermediate host. Since the source of this virus has not been identified yet recurrent reintroduction of the virus into the human population is causing new outbreaks worldwide.

As for all CoVs, the SARS-CoV-2 structure includes four main proteins that are important for virion assembly and infection: the spike glycoprotein (S), the envelope protein (E), the membrane protein (M), and the nucleocapside protein (NC), as well as 16 major nonstructural proteins.

As for SARS-CoV, the S-protein of SARS-CoV-2 is involved in the host tropism by attachment to ACE2, a membrane enzyme that is recognized as a functional receptor from the virus [8,11,14]. The ACE2 amino acid primary structure shows high sequence similarity in different animal species including humans, non-human primates, pets, domestic animals, and wild animals, with the risk of cross-species transmission [15].

Selected epitopes from the S of SARS-CoV-2 share high similarity with four epitope sequences of the S protein from canine respiratory coronavirus CoVBJ232, which is transmitted via direct contact between infected animals [16] and, to a lesser extent, with the bovine coronavirus and the human enteric coronavirus [17]. 

The nucleocapside (NC) is highly immunogenic and can induce significant antibody production in infected subjects, but less so in convalescent subjects. It is also involved in the replication, transcription, and packaging of its viral genome, as well as in hindering the reproductive cycle of the host cell [18]. Its sequence is highly conserved among species. The comparison of the whole proteic sequence showed that the NC from SARS-CoV-2 is similar to bat CV RaTG13 at above 99%, similar to SARS-CoV at about 90.3% and similar to pangolin CoV at almost 88%. In contrast, the NC proteins of bovine, canine, human enteric, and avian coronaviruses are the least similar [19]. Finally, the SARS-CoV-2 envelope (E) protein constitutes 75 amino acids existing in both monomeric and homo-pentameric forms. E proteins localize within the secretory pathways at the interspace between the endoplasmic reticulum and Golgi apparatus, to favor viral replication and dissemination within the host body. They are also immunogenic [20]. This suggests that a comparison of the full protein sequence of the E proteins from taxonomically related coronaviruses with tropism for the mostly synanthropic animals interacting with the global human population, would help in determining the common antigenic sites that would give partial protection from human–animal interactions. A comparison of the full E protein sequence reveals that the SARS-CoV-2 protein is 100% identical to the bat coronavirus and the pangolin coronavirus. Less than 95% sequence similarity is found with SARS-CoV and minor (40%) sequence similarity is observed with the causal agents of MERS, MERS-CoV, and the camel and dromedary coronavirus. There is a similarity slightly below 31% with the bovine and canine coronaviruses, and 20% with the chicken coronavirus. Although the human enteric CoV, bovine CoV, and canine CoV have a reduced sequence homology with SARS-CoV-2, they share several SARS-CoV-2 epitopes in their E protein that suggest a potential immunogenicity and their possible use in prophylactic-oriented investigations [20]. Thus, by comparing the epitopes of taxonomically related viruses, it is possible to determine an eventual cross-protection between humans and domestic animals due to the continuous inter-relations among them. 

Human-to-human transmission of SARS-CoV-2 mainly occurs when people are in close contact with people with COVID-19. The virus is primarily transmitted through droplets in the air that are generated when an infected person coughs, sneezes, or speaks, or via oro-fecal contamination [21]. It is possible to become infected by breathing in the virus within a 1 m distance from a person who has COVID-19. It has been reported that SARS-CoV-2 can survive on surfaces for up to 9 days [22]. Therefore, washing one’s hands properly before touching one’s eyes, nose, or mouth after touching a contaminated surface may reduce the risk of infection. People with COVID-19 develop symptoms after an incubation period that ranges from 1 to 14 days (commonly around five days). Symptoms include fever, coughing, dyspnoea, anosmia, agenousia, nausea, vomiting, diarrhea, sore throat, headache, and myalgia [22,23,24,25]. In more severe cases, patients can develop pneumonia. In the most severe cases, oxygen or ventilation are required. SARS-CoV-2 has primarily been detected in respiratory tract tissue, as well as in other tissues such as the intestines [26]. 

People of any age can be infected with SARS-CoV-2. The transmission chain can be initiated by asymptomatic [23,27,28,29,30] or mildly affected people with COVID-19 [31]. Older people and people with comorbidities, such as asthma, diabetes, heart disease, and cancer, tend to become severely ill, develop a hyperactive immune response and acute respiratory distress syndrome (ARDS) [21]. A viral-load-independent, different response to COVID-19 in healthy people has also been attributed to genetic predisposition, in particular, the prevalence of permissive HLA alleles that lead to an extreme and often lethal inflammatory reaction [32,33]. Two recent papers suggest that people with blood group O may have a decreased risk of contracting SARS-CoV-2 or may only exhibit mild symptoms [34,35]. More recently, as the pandemic expanded, women appeared to be affected just as frequently as men, representing about half of the global adult population. However, women appeared to suffer and die less frequently from COVID-19 than men. This was attributed to their different innate immunity, steroid hormones, and the properties of their sex chromosomes [36,37,38]. Infected children exhibit fewer clinical symptoms and might escape observation; however, they are still capable of transmitting the virus to the others [39,40]. Nevertheless, children can develop serious conditions, particularly when affected by basic diseases, long-term use of immunodepressants or when they are immunocompromised [41]. There does not seem to be any evidence of vertical transmission by an intrauterine infection in females who suffered from COVID-19 interstitial pneumonia in late pregnancy; the virus has not been found in amniotic fluid or breast milk [42,43]. 

Patients from different regions in Asia have been found to have been infected by different strains of SARS-CoV-2, which have displayed different virulence and epidemic behaviors. Based on their phylogenetic relationships, they have been divided into at least six different genotypes [44]. From December 2019 to April 2020, a comparative study of the complete genome sequences of 95 SARS-CoV-2 variants that were present in GenBank, the National Microbiology Data Centre (NMDC), and the NGDC (National Genomics Data Center) Genome Warehouse, demonstrated a strong association between sample collection time, location of the sample, and accumulation of genetic diversity. One hundred and sixteen mutations of the virus have also been found, suggesting the evolution of the virus and the coexistence of different strains in Asia, Europe, and North America [45], which might affect the severity and spread of SARS-CoV-2 [46]. 

While a few vaccines have been developed and are almost ready to be put on the market, several licensed medicines are now administered to reduce the symptoms. Lactoferrin, a natural iron-binding glycoprotein of the transferrin family, is secreted by glandular cells and is found in most bodily fluids. It occurs in mammalian milk [47] and was first identified in bovine milk [48]. Lactoferrin has a wide range of antiviral, immunomodulatory, and anti-inflammatory effects. In vitro studies have shown antiviral activity against a wide range of viruses, including SARS-CoV, that are closely related to SARS-CoV-2. It seems to be responsible for the absence of symptoms in infants born to mothers with COVID-19 [49,50]. Because of its immunomodulatory and anti-inflammatory activities, Lactoferrin has been suggested as a potential adjunct in severe COVID-19 cases [51,52].

Several measures, such as quarantining and social distancing, have been adopted to help interrupt the human-to-human chain of transmission to contain the COVID-19 outbreak [53]. The World Health Organization (WHO) has advised everyone to protect themselves from the virus by following good hand hygiene, wearing protective masks, and avoiding gatherings. During summer, the average age of affected people reduced, comprising those between 20 and 50 years old, due to the relaxation of restrictions on attending concerts, nightclubs, and travelling to epidemic ccountries. The number of infected teenagers also seemed to increase. As of 21 November 2020, 56,982,476 people have been infected by COVID-19 and 1,361,847 people have died worldwide [54]. 

Coronaviruses infect also different wild (bat, palm civet) and domestic animal species (dogs, cats, bovine) which show respiratory symptoms that can be accompanied by other symptoms such as enteritis [1,7]. Thus, these zoonotic causing agents are a constant threat to the public health.

Dogs and cats are often in close contact with humans. Isolated cases of pets that tested positive for COVID-19 after being in close contact with an affected human have been reported by the World Organization for Animal Health (OIE), an intergovernmental organization which coordinates, supports, and promotes animal disease control, and prevent their spread [55]. The first case was reported in February 2020, in Hong Kong, where a pet dog whose owner was affected by COVID-19 tested weak-positive, suggesting a potential human-to-animal infection. The dog had no symptoms but was quarantined at a government facility. Quantitative real-time polymerase chain reaction (RT-PCR) testing revealed the presence of SARS-CoV-2 RNA in the canine nasal swabs, while the rectal and fecal specimens were negative. The dog finally tested negative. In March 2020, a second dog whose owner had COVID-19 tested positive. Both the oral and nasal canine swabs collected tested positive, while rectal swabs tested positive in the first tests and negative subsequently [56]. In July 2020, a pet cat whose owner was affected by COVID-19 tested positive for SARS-CoV-2. The cat did not exhibit any specific clinical signs.

One study reported that ferrets and cats were sensitive to SARS-CoV-2 and cats were susceptible to transmission through the air. Dogs were less sensitive, and livestock (including pigs, chickens, and ducks) were virus-resistant [57]. Zhang et al. (2020) [58] looked at cats living in Wuhan before and after the outbreak. All cats were first tested for antibody reactivity against the recombinant receptor-binding domain (RDB) of the SARS-CoV-2 spike protein using an indirect enzyme-linked immunosorbent assay (ELISA). Later, the presence of SARS-CoV-2-specific neutralizing antibodies was determined by virus neutralization tests (VNT). Some of the cats had a strong SARS-CoV-2 antibody titer, and all of them had been in contact with people affected by COVID-19. The other cats in the study, which were from a pet hospital, displayed low neutralizing antibody titers that might have been due to the close contact between cats and COVID-19 affected patients. Finally, some stray cats were found to have been infected, which might have been due to contact with an infected surface or a person with COVID-19. Therefore, this study strongly excluded a possible SARS-CoV-2 transmission from cats to humans. Meantime, mildly symptomatic or asymptomatic dogs and cats owing to COVID-19 positive patients were reported from several countries. Thus, the OIE advises that people with COVID-19 must avoid contact with their pets. Basic hygiene measures should be taken when handling and taking care of animals. These include washing hands before and after touching the animals, their food, or their supplies, and avoiding kissing, licking, or sharing food. People with COVID-19 should ask other members of the family to care for their animals, which should be kept indoors as much as possible.

The golden Syrian hamster is an experimental animal model used for studying respiratory viruses such as SARS-CoV, the influenza virus, and the adenovirus [59,60]. Chan and coworkers [61] and Sia and coworkers [62] proposed the golden Syrian hamster as an experimental animal model for studying COVID-19 pathogenesis, antiviral treatment, and vaccine. The hamsters were infected with SARS-CoV-2 isolated from a nasopharyngeal aspirate specimen from a patient with COVID-19 in Hong Kong [61,62], and recapitulated the clinical, virological, histopathological, and immunological characteristics of human diseases [61]. More recently, the hamster has been introduced into families as a small pet, therefore it might also be infected if its owner has COVID-19.

Minks farms in the United States of America, Netherlands, Spain, Denmark, Sweden, Greece and Italy tested positive for COVID-19 [55]. Unfortunately, minks have been found very sensitive to SARS-CoV-2: in this species the virus progresses quickly, and most infected mink die the following day. Two variants of SARS-CoV-2 have been found in the mink farms COVID-19 positive in Netherlands and Denmark that were transmitted to mink farmers. One of these variants of the virus called “cluster 5” has been found in the Danish mink farm and has been transmitted to mink farmers. It shows four genetic changes, three substitutions and one deletion in the S protein. Since the S protein contains the receptor-binding domain and is one of the targets for immune response, such mutation might affect treatment, certain diagnostic tests and virus antigenicity, impacting the effectiveness of the running developing vaccine candidates, requiring them to be updated. Nevertheless, the probability of infection with mink-related variant strains was assessed as low for the general population, moderate for populations in areas with a high concentration of mink farms and very high only for individuals with occupational exposure [63]. This variant has not been reported from other countries, suggesting that it has not spread beyond Denmark. Animal species infections reported worldwide are shown in Figure 1.

Appropriate technical and organisational measures should be taken to ensure the health and safety of workers in the workplace, informing, training and providing them with appropriate personal protective equipment, including respiratory and eye protection. Testing of workers, contact racing, isolation, and quarantine to be immediately initiated when a human case is related to an animal farm in all cases of epidemics such as COVID-19.

Isolated SARS-CoV-2 strains genotyped systematically according to validated protocols and genome sequences from all infected animals—in particular mink—should be shared. This will enable the rapid identification of possible clusters and related variants.

Hunting and wildlife farms put humans in direct contact with numerous wildlife species used for legal or illegal consumption, or trade in wildlife markets around the world [64]. Due to cramped cages, poor biosecurity, and unhygienic handling of animal excreta, those animals suffer from debilitating and immunocompromising conditions that promote zoonotic outbreaks because disease surveillance of them is largely absent, as in the cases of SARS and Ebola outbreaks [65,66]. Li and coworkers [15] compared human-conserved ACE2 residues, the recognized receptor for the S region of SARS-CoV-2, with the ACE2 residues from different animal species and found a high sequence similarity, suggesting the risk of a potential interspecies transmission of SARS-CoV-2 among non-human primates, domestic, farm and wild animal species. Therefore, extending disease surveillance to different animal populations will be necessary. The Convention on the International Trade in Endangered Species (CITES) regulates international wildlife trade based on species’ endangered status, but only a few countries use strict veterinary import controls, and there are no global regulations on pathogen screening associated with the international trade in wildlife [65].

Determining the immunological properties of the circulating SARS-CoV-2, and which epitopes among coronaviruses with tropism for the most common synanthropic animals are co-shared, might also suggest a partial protection arising from human–animal interaction. It might provide a valuable list of epitopes that could be used for the differential diagnosis of SARS-CoV-2 infection [20].

Environmental contaminants such as heavy metals, doxins, PCB, and myco- and phytocotoxins are shared between humans, animals [67], and plants. Humans and animals share the same sources of food and water; thus, both are exposed to the same contaminants. In addition, humans can be contaminated through ingestion of contaminated animal products, and contaminated plants that are consumed as part of the human diet. Moreover, animals are sensitive indicators of environmental chemical hazards and may serve as sentinels for human environmental risks. 

By understanding the causes of disease emergence and the ecology of the agents involved (and their animal hosts), the creation of a network capable of merging the contributions of different areas of expertise might be possible. At present, medical doctors, veterinarians, public health experts, and food quality inspectors act separately. For example, veterinarians are not connected with occupational physicians, and the latter are not in contact with general practitioners who are at the frontline of the disease. 

In 2018, the WHO, Food and Agricultural Organization (FAO), and OIE formed a formal partnership to combat human–animal–environmental health risks. In 2019, they released a multisectoral guide with a One Health approach (1): a tripartite guide to addressing zoonotic diseases in countries that provides principles and best practices to assist countries in achieving sustainable and functional collaboration at human–animal–environmental interfaces. Such efforts should also be made at academic and public health levels. One Health should include a systematic surveillance of all zoonotic viruses in the different animal species as well as of professionals who have greater exposure to zoonotic risks, e.g., farmers or butchers. That would be essential for early warning and preparedness for the next any potential pandemic. Public health departments, hygienists, occupational physicians, general practitioners, and veterinarians should implement monitoring activities, where the health of workers, animals, and the general population would be jointly monitored. Governments should introduce the One Health approach in education and training programs and in their public health systems. Multidisciplinary cooperation in the implementation of programs, policies, and research would aim to meet the common goals of disease prediction, prevention, and medical care to be performed at local, regional, national, and global levels [68,69].

The concept of One Health recognizes the interdependence of human, animal, and environmental health, and aims to improve public health outcomes through the understanding and prevention of risks that originate at the interfaces between humans, animals, and their environments [68,69].

## 2. Conclusions

CoVs are a severe global health threat. Animals, humans, and plants are part of a unique ecosystem and should be treated as a One Health integrated organization [70]. New outbreaks may continue to be frequent in the future due to increased interactions between humans and animals, and changes in climate and ecology that could modify pathogen characteristics and cause molecular evolutions in different host species, as well as host responses to infections and interactions between wild animals, domestic animals, and humans, making prevention activities difficult.

The spread of COVID-19 in humans occurs mainly by human-to-human transmission. Among pet animals, cats (and to a lesser extent dogs) are sensitive to SARS-CoV-2 infection and occasionally display symptoms, but they do not infect humans. Therefore, there is a possibility that pets could contract COVID-19 through close contact with people that have COVID-19. Although it is rare, the possibility of SARS-CoV-2 transmission to pets should be considered to protect animals and the people who take care of them. In contrast, data collected so far seem to exclude pets from having an epidemiological role in the spread of the virus to humans. Thus, the abandonment of COVID-positive (or presumed positive) animals is unjustified and cruel. Minks, ferrets, and hamsters exhibit more dramatic symptoms. The finding of the cluster 5 in mink that can be transmitted to humans who have close contact with this species suggests the possible role of the mink as a reservoir of the virus. 

Although animals may play a critical role in the diffusion of epidemics as a virus reservoir and contribute to the evolution of some viruses, animals may also play a role as a beneficial source of immune-stimulation against similar viruses.

Thus, SARS-CoV-2 risk assessments of synanthropic animals should be carried out through clinical surveillance, seroprevalence, and other epidemiological and experimental research activities. It is important to understand the risk that people with COVID-19 pose to the animals with which they interact so that effective recommendations and risk management measures against COVID-19 can be made using a One Health approach.

## Figures and Tables

**Figure 1 animals-11-00016-f001:**
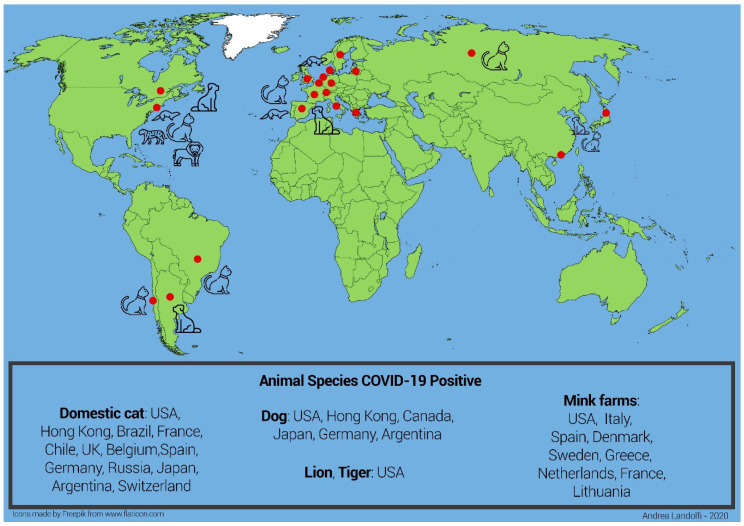
Graphical representation of the global epidemiology of COVID-19. The epidemic was first reported in Wuhan (red dot) in November 2019. From there it spread to all Asian provinces and Australia. At the beginning of 2020, the epidemic spread to Europe (with Italy first and other European countries following) and then Africa. From April 2020 the epidemic spread to the US. In 2020 a few cases of domestic animals (dog and cat), farm animals (mink) and wild animals (tiger and lion) being infected by their COVID-19-positive owners or local workers were reported in several countries (Countries: red dot; all animals: black color). Data from the OIE (Official International des Epizooties: The World Organization for Animal Health).

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
