# Peer review of "Do Animals Play a Role in the Transmission of Severe Acute Respiratory Syndrome Coronavirus-2 (SARS-CoV-2)? A Commentary"

_animals, 2020, doi:10.3390/ani11010016_

Round 1

Reviewer 1 Report

Firstly, the manuscript is improved as compared to the original submission. The point the authors are trying to make is worthwhile, i.e. that One Health is important. However, as it stands this point is not particularly obvious. A much better approach would have been to have a manuscript reviewing One Health, with COVID as one of a variety of examples. 

Unfortunately, I still feel that this work is not of sufficient quality to be published without substantial further editing. There has been an appreciable amount of editing, but the English remains poor. The figure is largely unchanged, and unfortunately I still feel it is of little value in this manuscript.

There are also some puzzling sentences (possibly grammar related), for example "...animals may also play a role as beneficial source of immune-stimulation against similar viruses."

In summary, I feel that this paper would be better written as a One Health paper rather than a COVID-19 paper.

Author Response

First of all, we wish to thank reviewers 1, 2 and 3 for helpful criticism and suggestions that allowed us to improve our manuscript.

As suggested, we have extended in a “One Health” vision deepening the role of the animals in the current pandemic but, although knowledge of SARS Cov-2 has progressed, there is still no scientific evidence yet that animals can transmit the virus to humans, as result of the ability of the present Sars CoV-2 transmitted from human mainly to cat to return to humans. Minks are very sensitive to the virus.  This event has caused that in several countries such as US, Denmark, Spain, Netherlands, many mink farms close to farms or herds confirmed or suspected to be infected with COVID-19 have been culled with an enormous economic damage to the farmers and general economy of the countries involved. According to the Danish government a SARS-Cov-2 variant related to minks has been found in humans. Thus, the probability of infection with mink-related variant strains is considered as low for the general population, moderate for populations in areas with a high concentration of mink farms and very high for individuals with occupational exposure. Patients reported to be infected with mink-related variants, including the Cluster 5 variant in Denmark, do not appear to have more severe clinical symptoms than those infected with non-mink-related variants. Therefore, the current impact of COVID-19 on disease severity in patients infected with any mink-related variant appears to be similar to those infected with non-mink-related variants. This impact was previously assessed as low for the general population and very high for individuals with risk factors for severe COVID-19 disease, such as the elderly (European Centre for Disease Prevention and Control. Detection of new SARS-CoV-2 variants related to mink – 12 November 2020. ECDC: Stockholm; 2020).

As suggested, the entire manuscript has been enriched with matters and relative references. The manuscript was edited in the first version by Elsevier Editing Service and in this one, by MDPI English editing.

Reviewer 2 Report

The revised version of the manuscript written by Costagliola and colleagues has been improved according to the Reviewer’s suggestion. Although criticisms have been amended, several imprecision and grammar errors have been highlighted throughout the text, reducing manuscript fluency and understandability. Apart from these aspects, the manuscript is, to this Reviewer, suitable for publication.

Author Response

Imprecision and grammar errors have been corrected by supervision of the MDPI English editing.

Reviewer 3 Report

The title of the manuscript addresses a very interesting issue in the COVID-19 disease, i.e. the possible role of animals (including both domestic and wild animals) in the transmission of the virus. Despite the interesting idea, however, the commentary is a rather unfocused summary of the COVID-19 story, describing symptoms, viral proteins, even including possible treatments, while the role of animals is just briefly and insufficiently touched. As such, the manuscript is superficial and essentially useless. I would suggest, should the authors decide so, to re-propose a real commentary on the role of animals in the viral transmission, as the title was promising.

Author Response

As suggested, we have extended and deep the concept of One Health throughout the manuscript. We aimed to focus on the possible role of the animals in the COVID-19 pandemic that differs from other forms of zoonosis and the role that the veterinarians play within the WHO system.

Round 2

Reviewer 1 Report

Firstly, there is a clear improvement from the original draft. However, I still feel that the English is occasionally poor. Similarly, the figure has not really been improved; the outline animals really don't add much. Why not use different colours for different species, for example? Ultimately a table would suffice for these data.

The manuscript would benefit from carefully restructuring to provide a clearer narrative. Currently it reads like a long commentary that doesn't always flow logically. For example, lines 282-286 suddenly introduce influenza, before switching to ACE2 and COVID-19. There is some mention of pigs, but then there is suddenly a discussion of wildlife.

A previously mentioned, I feel that this manuscript suffers from being in a crowded arena. There are many similar papers discussing this work and this manuscript struggles in comparison. Furthermore, the pace of change is so rapid that a COVID-specific review is almost immediately out of date. Hence, I feel that a more general review would stand a better chance.

Author Response

First of all, we wish to thank reviewers for helpful criticism and suggestions that allowed us to improve our manuscript.

Firstly, there is a clear improvement from the original draft. However, I still feel that the English is occasionally poor.

As suggested,  the manuscript was edited by MDPI English editing.

Similarly, the figure has not really been improved; the outline animals really don't add much. Why not use different colours for different species, for example? Ultimately a table would suffice for these data.

As suggested, we have modified the Figure

The manuscript would benefit from carefully restructuring to provide a clearer narrative. Currently it reads like a long commentary that doesn't always flow logically. For example, lines 282-286 suddenly introduce influenza, before switching to ACE2 and COVID-19. There is some mention of pigs, but then there is suddenly a discussion of wildlife. A previously mentioned, I feel that this manuscript suffers from being in a crowded arena. There are many similar papers discussing this work and this manuscript struggles in comparison. Furthermore, the pace of change is so rapid that a COVID-specific review is almost immediately out of date. Hence, I feel that a more general review would stand a better chance.

As suggested, we have restructured the manuscript.

Reviewer 3 Report

The authors have significantly improved the manuscript, although some parts are still not fully within the scopes of the study.

Author Response

First of all, we wish to thank reviewers for helpful criticism and suggestions that allowed us to improve our manuscript.

First of all, we wish to thank reviewers for helpful criticism and suggestions that allowed us to improve our manuscript.

As suggested, we have deleted some sentences not fully within the scopes of the study.

This manuscript is a resubmission of an earlier submission. The following is a list of the peer review reports and author responses from that submission.

Round 1

Reviewer 1 Report

The manuscript of Costagoliola et al is a useful commentary on the role of animals in the current COVID19 pandemic as means of the spread of the SARSCoV2. To this Reviewer opinion the manuscript lack to consider several aspects of the role animals in the current pandemic as, for instance, the potential role the diverse animal species may have (proved or supposed) in the virus life cycle. Sharing of the etiopathogenetic features between human and animal hosts are only barely touched in the manuscript and, to this Reviewer opinion, these aspects have to be comprehensively elucidated while talking of One Health concept.  Also, in the context of the epidemiological importance of the animals author did not mention previous studies where, using a one health concept, immunological properties of the circulating and previous animal CoV have been compared. Highlighting similar immunological features is of importance for the whole epidemiology of the virus as the owners of animals with CoV infections might be responsive to the SARSCoV2 (doi: 10.1016/j.micinf.2020.05.013; doi: 10.1016/j.micinf.2020.03.002; https://doi.org/10.1016/j.micinf.2020.04.002). Finally, the wishes of the authors (i.e. continuous dialogue between the human and veterinary medicine) are not exhaustively and properly supported in the manuscript. Although this Reviewer agrees with the authors' view, strengths of the constant dialogue between the human and veterinary medicine have to be emphasised by providing practical examples that make this concept clear to a much wider audience.

Reviewer 2 Report

The authors should be congratulated on identifying an important concept in the ongoing outbreak of SARS-CoV-2, namely the role which domestic animals play in the epidemiology of the disease.

However, I believe there are a variety of factors which make this manuscript unpublishable in its current form.

For a field which is so dynamic and richly referenced there are sections of the manuscript that are poorly referenced - if at all. This also relates to a key point regarding the manuscript: the title claims to be a 'systematic review'. Systematic reviews are, by definition, systematic and follow a pre-determined, set protocol. There is no such protocol described here. Instead the paper reads more like a mini-review of various reports of animals becoming infected with SARS-CoV-2.

In my opinion the figure doesn't add anything. Is there a point to displaying this information on a map, particularly in the absence of any explanatory data, for example the incidence of human cases?

In summary, I feel that the manuscript would be greatly improved by not claiming to be a systematic review, being shortened and improving the English. A veterinary - specific journal might also be a better fit.